# Statistically Calibrated Scaling for Token Merging in Transformers

**Qing Zhou** [1]   **Hongyuan Zhang** [2]   **Tao Yang** [1]   **Junyu Gao** [1]   **Qi Wang** [1]

*Figure 1.* Left: Our $\sqrt{r}$-scaling mitigates the sharp accuracy degradation of baselines (Vanilla/PA), recovering +29% accuracy at aggressive compression. Middle: Compared with the radial drift observed in PA, our method constrains energy inflation. Right: Our approach improves angular fidelity (higher cosine similarity), preserving feature semantics under heterogeneous merging.

## Abstract

Token merging accelerates Transformer inference by clustering similar tokens to reduce sequence length (retention ratio $r$), but it can distort attention outputs, inducing covariate shift in residual streams and performance degradation under high compression. Existing heuristics, such as proportional attention, work well under mild compression but degrade at aggressive ratios due to uncorrected energy drift and biased attention distributions. We reframe token merging as a statistical reconstruction problem in high dimensions and introduce an asymptotic radial-angular decomposition of the reconstruction error, separating magnitude and distributional distortions. Under finite second-moment and variance stationarity assumptions, minimizing the leading-order decomposed risk yields closed-form corrections governed by a single factor $\sqrt{r}$: scaling merged values and shrinking merged logits toward the cluster-size prior. This provides a parameter-free calibration of energy balance and distributional fidelity. Extensive experiments on

vision Transformers demonstrate improved accuracy and robustness across reduction mechanisms and compression levels. Code: `https://github.com/mrazhou/sqrt_r`

## 1. Introduction

The quadratic complexity of self-attention (Vaswani et al., 2017) remains the primary bottleneck for deploying Vision Transformers (ViTs) (Dosovitskiy et al., 2021; Touvron et al., 2021; Liu et al., 2021) in resource-constrained environments. To circumvent this, token merging (Bolya et al., 2023) has been established as a powerful training-free paradigm, progressively reducing sequence length by averaging similar tokens, distinct from earlier pruning approaches (Rao et al., 2021; Liang et al., 2022; Yin et al., 2022). While effective in mild compression regimes, token merging methods can suffer from sharp performance degradation under aggressive reduction ratios (Figure 1). This degradation is not merely a consequence of information loss, but also reflects a systematic statistical misalignment in the residual stream, i.e., the main feature pathway accumulated through Transformer skip connections. Such misalignment induces covariate shift (Shimodaira, 2000) and can reduce the effectiveness of pre-trained weights under high compression.

Current mitigation strategies rely predominantly on heuristics, most notably proportional attention (PA) (Bolya et al., 2023), which injects a logarithmic bias $\ln N_j$ to correct cluster probability mass. This zeroth-order adjustment is

[1]School of Artificial Intelligence, OPtics and ElectroNics (iOPEN), Northwestern Polytechnical University, Xi'an, China. [2]University of Hong Kong. Correspondence to: Qi Wang <crabwq@gmail.com>.

*Proceedings of the 43rd International Conference on Machine Learning*, Seoul, South Korea. PMLR 306, 2026. Copyright 2026 by the author(s).

effective at low compression but can become insufficient in high-compression regimes. PA treats merged representations as deterministic summaries of their clusters, while leaving aggregation-induced variance dynamics largely uncalibrated. Specifically, Figure 1 illustrates two complementary distortion mechanisms: (1) *energy drift* in value vectors caused by changes in the summation degrees of freedom, and (2) *overconfidence* in attention logits (Guo et al., 2017) derived from noisy, heterogeneous clusters. As the retention ratio decreases, these uncorrected second-order effects can accumulate across layers and destabilize signal propagation (Dong et al., 2021).

In this work, we reframe token merging as a statistical reconstruction problem. Leveraging high-dimensional concentration arguments, we introduce an *asymptotic radial-angular decomposition* of the reconstruction error. This analytical framework separates distortions into radial components, corresponding to magnitude or energy drift, and angular components, corresponding to distributional mismatch, enabling the two effects to be analyzed and corrected independently at leading order.

Minimizing this leading-order decomposed risk under finite second-moment and approximate variance-stationarity assumptions yields a simple closed-form correction: a single parameter-free factor $\sqrt{r}$ governs both components of the surrogate objective. In particular, this factor calibrates energy dynamics by scaling merged values and mitigates distributional bias by shrinking merged logits toward the cluster-size prior with reliability weight $\sqrt{r}$. This unified correction, which we term $\sqrt{r}$-*scaling*, provides a statistically grounded calibration of the residual stream and improves robustness at high compression ratios where conventional heuristics often degrade. Extensive experiments on Vision Transformers demonstrate that our approach consistently improves accuracy and stability across a wide range of token reduction mechanisms and compression levels.

## 2. Related Work

**Efficient Vision Transformers via Structural Reduction.** The quadratic complexity of self-attention has spurred distinct lines of research in sequence reduction. Token pruning methods discard tokens deemed uninformative based on attention scores (Rao et al., 2021), or adaptive halting mechanisms (Meng et al., 2022; Yin et al., 2022). While some approaches like ATS (Fayyaz et al., 2022) and SiT (Zong et al., 2022) refine the selection policy to minimize information loss, pruning fundamentally removes part of the token sequence and may discard unrecoverable signal. Token merging addresses this limitation by aggregating redundant tokens rather than removing them. Following the bipartite matching paradigm of ToMe (Bolya et al., 2023), subsequent works have explored more sophisticated reduction

criteria. For instance, DiffRate (Chen et al., 2023) and ToFu (Kim et al., 2024) focus on differentiable compression and fusing pruning with merging, respectively. Meanwhile, DCT (He et al., 2023) and PiToMe (Tran et al., 2024) exploit spectral properties to preserve informative signals. These methods are also complementary to weight-side compression techniques, such as quantization (Nagel et al., 2021), since token reduction dynamically reduces sequence length while weight compression reduces model parameters or arithmetic precision. Despite these algorithmic diversities, prior token reduction works mainly focus on which tokens to keep, prune, or merge. In contrast, our work studies how the aggregation operation itself changes feature norms and attention logits, and proposes a lightweight calibration to reduce the resulting statistical shift.

**Statistical Consistency in Deep Signal Propagation.** Theoretical analyses of deep networks emphasize the critical role of variance preservation for trainability and inference stability. Foundational works on initialization (He et al., 2015) and normalization (Ba et al., 2016; Xiong et al., 2020) establish that feature activations should maintain controlled moments to prevent signal explosion or vanishing. Token merging introduces a dynamic, input-dependent change in sequence length from $N$ to $rN$, which can perturb these moment statistics even when the underlying pretrained model is stable. While intra-cluster averaging reduces the variance of individual merged features, the subsequent attention operation aggregates probability mass into fewer super-tokens and can introduce a counteracting energy drift in the residual stream. This phenomenon is related in spirit to variance shifts studied in quantization (Nagel et al., 2021) or domain adaptation, but differs because the distortion is induced online by input-dependent token aggregation. Our analysis connects this dynamic reduction process to a simple $\sqrt{r}$ calibration rule that approximately restores the statistical scale expected by the pretrained Transformer.

**Heuristic Corrections vs. Statistical Calibration.** Acknowledging the distortion caused by merging, heuristic corrections have been proposed. Most notably, Proportional Attention (Bolya et al., 2023) adds a logarithmic bias ($\ln N_j$) to merged logits to account for cluster size. From a statistical estimator perspective, PA performs a zeroth-order correction of cluster probability mass, but does not explicitly calibrate second-order effects such as value magnitude drift and logit reliability under heterogeneous merging. As we show in Section 3, these effects become more pronounced under aggressive compression. Our approach complements existing token selection or merging strategies by calibrating the continuous statistics of the merged representation. Under finite second-moment and approximate variance-stationarity assumptions, the resulting $\sqrt{r}$ factor arises as a closed-form leading-order correction for both radial energy calibration and angular logit shrinkage, without introducing additional

trainable parameters or empirical tuning.

# 3. Statistically Calibrated Scaling

Token merging accelerates Transformer inference by reducing the sequence length from $N$ to $M = rN$ ($r \in (0, 1]$), but it also distorts self-attention outputs. This distortion propagates through residual connections as covariate shift, degrading performance in deep models, particularly under aggressive compression ratios that require heterogeneous token grouping. We reframe token merging as a statistical reconstruction problem. By minimizing a leading-order approximation of the expected mean squared reconstruction error, we derive closed-form corrections that jointly stabilize magnitude and attention distribution under the stated assumptions.

## 3.1. Problem Setup

We consider a self-attention layer in a Transformer with embedding dimension $d$. Let the input token representations be $\mathbf{X} \in \mathbb{R}^{N \times d}$, from which queries $\mathbf{q}$, keys $\mathbf{k}_i$, and values $\mathbf{v}_i$ are linearly projected. The uncompressed attention output for a given query is

$$\mathbf{O} = \sum_{i=1}^{N} p_i \mathbf{v}_i, \quad p_i = \text{softmax}(z_i)_i, \quad z_i = \frac{\mathbf{q}^\top \mathbf{k}_i}{\sqrt{d}}. \quad (1)$$

Token merging partitions tokens into $M = rN$ clusters $\{\mathcal{C}_j\}_{j=1}^{M}$ with sizes $N_j = |\mathcal{C}_j|$, producing merged representations via intra-cluster averaging:

$$\mathbf{v}'_j = \frac{1}{N_j} \sum_{i \in \mathcal{C}_j} \mathbf{v}_i, \quad \bar{z}_j = \frac{1}{N_j} \sum_{i \in \mathcal{C}_j} z_i. \quad (2)$$

In pre-normalized Transformer blocks, the averaged residual token is normalized before key/value projection; thus, $\mathbf{v}'_j$ and $\bar{z}_j$ denote the projected quantities associated with the normalized merged token, while Eq. (2) only tracks the linearized averaging operation used to form cluster membership. This motivates our approximate variance-stationarity assumption, rather than an exact equality.

We parameterize corrections with value scaling $\alpha > 0$ and logit transformation $\mathcal{T}(\bar{z}_j, N_j)$, yielding the merged output

$$\mathbf{O}' = \sum_{j=1}^{M} \tilde{p}_j (\alpha \mathbf{v}'_j), \quad \tilde{p}_j = \text{softmax}(\mathcal{T}(\bar{z}_j, N_j))_j. \quad (3)$$

Our objective is to minimize the expected reconstruction error in the residual stream:

$$\min_{\alpha, \mathcal{T}} \mathcal{E}(\alpha, \mathcal{T}) = \mathbb{E}\left[\|\mathbf{O}' - \mathbf{O}\|^2\right], \quad (4)$$

where the expectation is over data and clustering. The analysis proceeds under finite second-moment and approximate variance-stationarity assumptions, detailed in Appendix A.1.

## 3.2. Asymptotic Decomposition of Reconstruction Error

Direct minimization of Eq. (4) is analytically intractable. In high-dimensional Euclidean spaces, however, concentration effects make magnitude and direction approximately separable to leading order. This yields the following asymptotic decomposition of the reconstruction error (Lemma 3.1):

$$\mathcal{E} \approx \underbrace{\mathbb{E}\left[(\|\mathbf{O}'\| - \|\mathbf{O}\|)^2\right]}_{\mathcal{E}_{\text{radial}} \text{ (Energy Drift)}}$$
$$+ \underbrace{2\mathbb{E}[\|\mathbf{O}\|^2] \cdot \mathbb{E}\left[1 - \cos(\mathbf{O}', \mathbf{O})\right]}_{\mathcal{E}_{\text{angular}} \text{ (Distribution Shift)}} + o(1). \quad (5)$$

The radial term captures magnitude drift arising from aggregation-induced variance changes in the weighted sum. The angular term reflects distributional mismatch caused by intra-cluster logit smoothing and heterogeneous token grouping. This decomposition provides a tractable leading-order surrogate in which the two effects can be analyzed and corrected separately.

**Lemma 3.1** (Asymptotic Radial-Angular Decomposition)**.** *Under finite second moments and high-dimensional concentration, magnitude fluctuations concentrate around their expectations and the radial-angular cross-covariance vanishes asymptotically. Consequently, the reconstruction risk admits the leading-order decomposition $\mathcal{E} = \mathcal{E}_{radial} + \mathcal{E}_{angular} + o(1)$.*

## 3.3. Limitations of Heuristic Scaling

PA relies on the logit bias $\mathcal{T}(\bar{z}_j, N_j) = \bar{z}_j + \ln N_j$ but neglects value scaling ($\alpha = 1$). While this zeroth-order correction is effective under mild compression, it can become insufficient when the retention ratio is low and clusters become heterogeneous. We analyze these limitations through the radial-angular decomposition in Eq. (5).

**Radial Drift: Neglect of Aggregation-Induced Energy Drift.** PA applies no correction to value vectors. The attention output is a weighted sum of random variables with finite second moments. Merging reduces the number of summands from $N$ to $rN$, altering the degrees of freedom in the summation. When attention mass is aggregated over clusters of average size $k \approx 1/r$, the squared cluster mass introduces positive cross-terms. In homogeneous or dispersed-attention regimes, this induces an inflation pressure that can scale up to a factor of $1/r$ in the expected output energy (Appendix A.3):

$$\frac{\mathbb{E}[\|\mathbf{O}'_{\text{PA}}\|^2]}{\mathbb{E}[\|\mathbf{O}\|^2]} \lesssim \frac{1}{r},$$

with the upper-bound behavior approached when within-cluster attention weights are sufficiently homogeneous. By neglecting $\mathcal{E}_{\text{radial}}$, PA leaves this energy drift uncalibrated in the residual stream.

**Angular Bias: Insufficient Compensation for Intra-Cluster Heterogeneity.** PA treats the averaged cluster logit $\bar{z}_j$ as an estimator of the cluster's log-probability mass, corresponding to a zeroth-order approximation of the Log-Sum-Exp function. This approximation incurs a second-order residual proportional to the intra-cluster logit variance $\sigma_j^2$ (Appendix A.5):

$$\mathbb{E}\left[\ln\left(\sum_{i\in\mathcal{C}_j}e^{z_i}\right)-(\bar{z}_j+\ln N_j)\right]\approx\frac{1}{2}\sigma_j^2.$$

As compression becomes more aggressive, clusters tend to encompass more heterogeneous tokens, causing $\sigma_j^2$ to increase. PA therefore does not explicitly account for the reduced reliability of noisy merged logits, which can induce attention misalignment and only partially mitigate $\mathcal{E}_{\text{angular}}$.

Together, the absence of radial calibration and the incomplete modeling of angular reliability explain why PA can degrade under aggressive compression.

### 3.4. Closed-Form Statistical Corrections

We now derive closed-form corrections for $\mathcal{T}$ and $\alpha$ under the leading-order surrogate objective. Proofs and implementation are detailed in Appendix A.3–A.4 and B, respectively.

**Radial Correction: Counteracting Aggregation Inflation.** The expected energy of the attention output is governed by the second moment of the attention weights, i.e., the inverse participation ratio: $\mathbb{E}[\|\mathbf{O}\|^2]\propto\mathbb{E}[\sum p_i^2]$. Token merging aggregates probabilities from a cluster of size $k\approx 1/r$ into a single mass: $P_j=\sum_{i\in\mathcal{C}_j}p_i$. Due to the convexity of the square function, the squared mass includes positive cross-terms:

$$P_j^2=\left(\sum_{i\in\mathcal{C}_j}p_i\right)^2=\sum_{i\in\mathcal{C}_j}p_i^2+\underbrace{\sum_{i\neq l}p_ip_l}_{\text{Cross-terms}}. \qquad (6)$$

In the homogeneous-cluster regime, these cross-terms cause the expected second moment to scale approximately linearly with the cluster size $k=1/r$, yielding an upper-bound inflation factor of $1/r$. To neutralize this aggregation-induced inflation pressure, the scaling factor $\alpha$ applied to the value vectors should satisfy the equilibrium condition:

$$\alpha^2\cdot(1/r)=1. \qquad (7)$$

Solving for $\alpha$ yields the radial correction:

$$\boxed{\alpha=\sqrt{r}.} \qquad (8)$$

This scaling should be interpreted as a conservative energy calibration: it cancels the combinatorial inflation component, while heterogeneous clusters may still exhibit benign deflation due to destructive interference.

**Angular Correction: LMMSE Estimation.** We optimize the logit transformation using Linear Minimum Mean Square Error (LMMSE) estimation. Under the standardized exchangeable cluster model in Appendix A.4, the merged cluster logit is treated as a normalized proxy for token-level logits. As the average cluster size grows to $k\approx 1/r$, this proxy becomes less reliable for representing individual token-level logit variation under heterogeneous grouping. The resulting reliability coefficient scales as

$$\rho\approx\frac{1}{\sqrt{k}}=\sqrt{r}. \qquad (9)$$

The LMMSE estimator therefore shrinks the merged logit $\bar{z}_j$ toward the geometric cluster-size prior $\ln N_j$, weighted by its reliability $\rho$:

$$\hat{z}_{\text{LMMSE}}=\rho\cdot\bar{z}_j+(1-\rho)\cdot\ln N_j. \qquad (10)$$

Substituting $\rho=\sqrt{r}$ gives

$$\boxed{\mathcal{T}(\bar{z}_j,N_j)=\sqrt{r}\,\bar{z}_j+(1-\sqrt{r})\ln N_j.} \qquad (11)$$

Intuitively, the shrinkage factor $\sqrt{r}$ acts as an adaptive temperature-like calibration. By damping noisy merged logits, it counteracts the overconfidence bias induced by intra-cluster variance (see Appendix A.6) and helps prevent the entropic collapse observed in PA (Figure 7).

**Theorem 3.2** (Leading-Order Calibration under Stationarity). *Under finite second-moment, approximate variance-stationarity, and the standardized exchangeable cluster model, the scaling pair $(\alpha=\sqrt{r},\mathcal{T}(\bar{z}_j,N_j)=\sqrt{r}\bar{z}_j+(1-\sqrt{r})\ln N_j)$ is the leading-order solution of the radial upper-bound calibration and angular LMMSE shrinkage surrogate induced by the radial-angular decomposition. The factor $\sqrt{r}$ calibrates radial energy and serves as a conservative reliability weight for the angular component.*

## 4. Experiments

We evaluate our $\sqrt{r}$-scaling across diverse reduction paradigms (merging, pruning, hybrid), model architectures, and multi-modal tasks. Our goal is to examine whether correcting statistical misalignment can mitigate performance degradation and maintain feature fidelity across different token reduction heuristics.

### 4.1. Experimental Setup

We compare our method against two primary baselines: Vanilla and PA, and further validate its generality by integrating it with diverse reduction mechanisms, including DCT, DiffRate, PiToMe, ToMe and ToFu. Experiments cover: (1) Image Classification on ImageNet-1K (Deng et al., 2009) using ViT; (2) Image-Text Retrieval on MS-COCO (Chen et al., 2015) and Flickr30k (Plummer et al.,

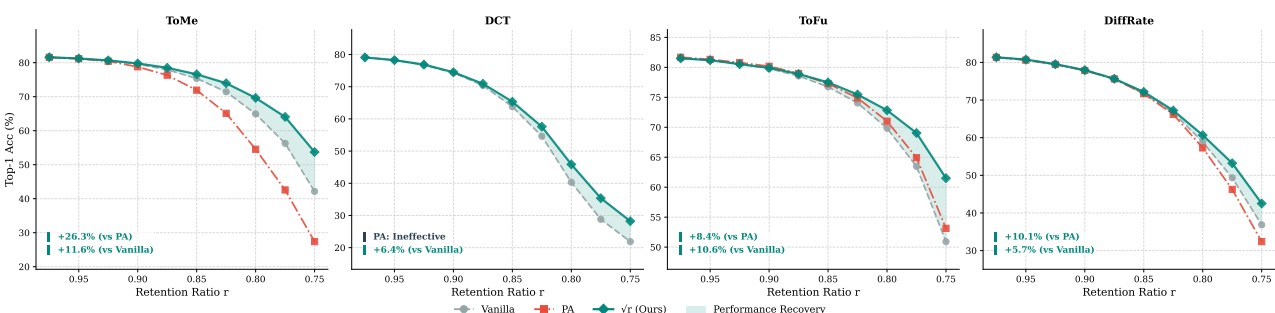

*Figure 2.* ImageNet-1K Classification Accuracy. We compare scaling strategies across different token reduction algorithms (ToMe, DCT, ToFu, DiffRate). As $r$ decreases, baseline methods (Vanilla, PA) exhibit **accelerated performance degradation**. In contrast, our $\sqrt{r}$-scaling (green) maintains a significantly flatter trajectory, with the performance gap widening progressively as compression intensifies.

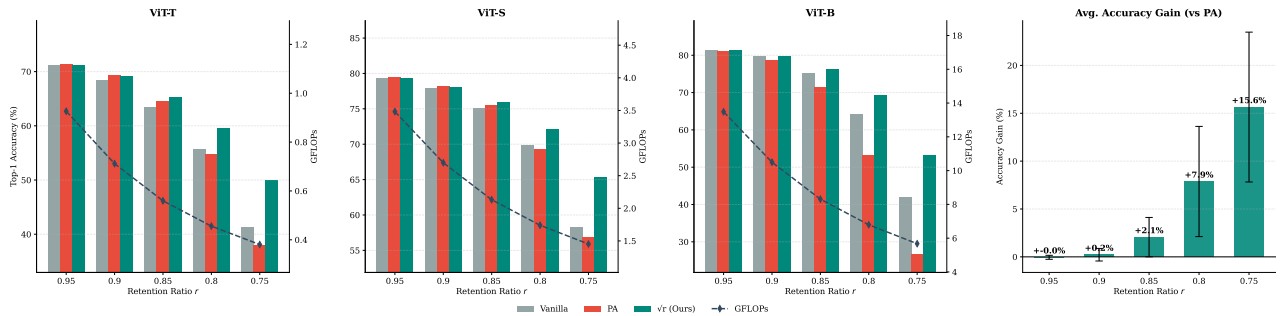

*Figure 3.* Scalability across Model Sizes with PiToMe. We evaluate the impact of scaling strategies using PiToMe (a spectrum-preserving merging method) as the base algorithm. Crucially, all scaling strategies share the same linear reduction in computational cost (GFLOPs, dashed line). Within this identical efficiency budget, the accuracy advantage of our method over baselines amplifies as the model size increases (from ViT-T to ViT-B) and the retention ratio decreases, reaching a maximum gain at the most aggressive setting ($r = 0.75$).

2015) using ALBEF (Li et al., 2021) and BLIP (Li et al., 2023a; 2022); (3) Visual Question Answering (VQA) on multiple benchmarks (GQA (Hudson & Manning, 2019), MME (Fu et al., 2023), etc (Gurari et al., 2018; Kembhavi et al., 2016; Li et al., 2023b; Yue et al., 2024; Mathew et al., 2022)).) using LLaVA-1.5 (Liu et al., 2024).

## 4.2. Consistency across Scales and Mechanisms

The central hypothesis of this work is that the rapid performance decay observed in existing token reduction methods is partly driven by cumulative statistical drift, beyond the unavoidable information loss caused by reducing tokens. We validate this consistency on ImageNet-1K classification, analyzing the degradation trends across compression ratios, model scales, and reduction algorithms.

**Mitigating Progressive Performance Degradation.** Performance trajectories diverge as retention $r$ drops (Figure 2). Baselines suffer accelerated decay: PA falls below the uncorrected Vanilla baseline at aggressive compression ($r = 0.75$). This inversion highlights the limitations of PA's heuristic, where zeroth-order estimation bias can become detrimental in heterogeneous clusters. In contrast, our $\sqrt{r}$-scaling stabilizes the trajectory by calibrating variance

dynamics. Acting as a statistical anchor, it maintains a flatter decay curve, widening the performance gap at $r = 0.75$.

**Scalability and Computational Efficiency.** Figure 3 reveals that statistical calibration becomes increasingly important as model capacity grows. The performance advantage of our method expands from ViT-Tiny to ViT-Base. In deeper networks like ViT-Base, the residual stream undergoes more aggregation steps, allowing unmitigated statistical errors such as radial drift to accumulate across layers. Our method acts as a layer-wise stabilizer that reduces this accumulation. Crucially, the dashed line in Figure 3 confirms that all three scaling strategies share identical GFLOPs reduction. Since our correction relies solely on scalar operations at inference time, it adds negligible computational overhead over Vanilla/PA, achieving considerable accuracy gains while retaining the same efficiency.

**Generality: From Merging to Hybrid Reduction.** Our correction is motivated by the statistical effects of reducing $N$ tokens to $rN$ tokens, making it largely agnostic to the specific selection heuristic. We verify this across diverse paradigms: bipartite matching (ToMe), spectrum-preserving merging (PiToMe), and differentiable selection (DiffRate). In all cases (Figure 2 and 3), $\sqrt{r}$-scaling consistently outper-

*Table 1.* Quantitative comparison on Image-Text Retrieval. We evaluate PiToMe with different scaling strategies on Flickr30k and MS COCO. Our $\sqrt{r}$-scaling consistently achieves the highest Recall Sum ($R_{sum}$), demonstrating superior feature alignment preservation.

| Dataset | Model | Ratio | Scaling | GFLOPs | Time | Text Retrieval | | | Image Retrieval | | | $R_{sum}$ |
|---|---|---|---|---|---|---|---|---|---|---|---|---|
| | | | | | | R@1 | R@5 | R@10 | R@1 | R@5 | R@10 | |
| Flickr30k | ALBEF | 1.0 | - | 65.5 | 2m59s | 95.8 | 99.8 | 100.0 | 85.5 | 97.4 | 98.9 | 577.4 |
| | | 0.90 | Vanilla | 36.5 | 1m38s | 86.9 | 95.7 | 98.4 | 73.4 | 91.8 | 94.7 | 540.9 |
| | | | PA | 36.5 | 1m40s | 86.2 | 95.4 | 97.3 | 72.0 | 90.2 | 93.8 | 534.8 |
| | | | $\sqrt{r}$ **(Ours)** | 36.5 | 1m37s | **89.0** | **97.1** | **98.7** | **75.5** | **93.0** | **95.9** | **549.1** |
| | BLIP | 1.0 | - | 65.5 | 48s | 96.9 | 99.8 | 100.0 | 87.5 | 97.6 | 99.0 | 580.8 |
| | | 0.90 | Vanilla | 36.5 | 24s | 92.7 | 98.6 | 99.6 | 80.5 | 95.6 | 97.7 | 564.7 |
| | | | PA | 36.5 | 24s | 92.4 | 98.2 | 99.1 | 79.6 | 94.6 | 97.2 | 561.1 |
| | | | $\sqrt{r}$ **(Ours)** | 36.5 | 24s | **93.4** | **99.0** | **99.5** | **82.1** | **95.9** | **97.8** | **567.8** |
| MS COCO | BLIP2 | 1.0 | - | 900.4 | 32m59s | 85.4 | 97.0 | 98.5 | 68.3 | 87.7 | 92.6 | 529.5 |
| | | 0.95 | Vanilla | 391.8 | 13m20s | 78.0 | 93.3 | 96.6 | 61.3 | 83.2 | 89.4 | 501.8 |
| | | | PA | 391.8 | 13m30s | 75.8 | 92.2 | 95.8 | 60.1 | 82.5 | 89.0 | 495.4 |
| | | | $\sqrt{r}$ **(Ours)** | 391.8 | 13m22s | **78.3** | **94.0** | **96.7** | **61.5** | **83.6** | **89.9** | **504.0** |

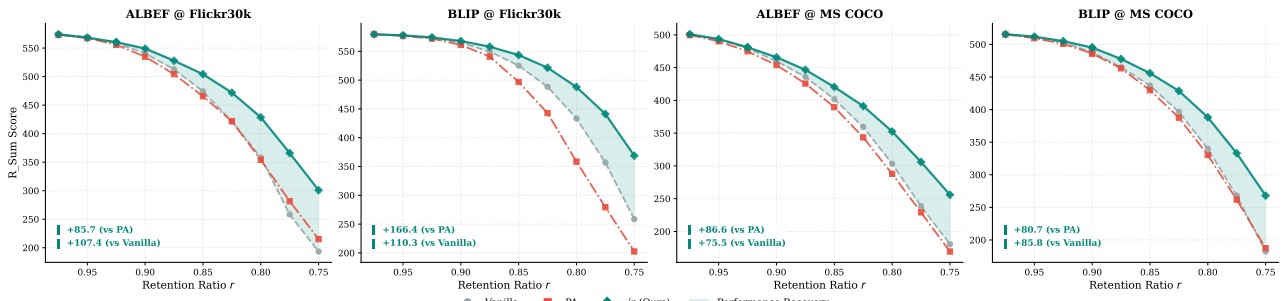

*Figure 4.* Retrieval Performance vs. Compression Ratio. Comparison of $R_{sum}$ scores on Flickr30k and MS COCO. Our method (green line) exhibits the slowest decay rate, highlighting its stability in preserving cross-modal alignment.

forms baselines, suggesting that statistical drift is a common consequence of token reduction rather than an artifact of a specific algorithm. We further extend this validation to complex architectures. First, for ToFu, which fuses pruning and merging, the variance shift persists despite its sophisticated selection strategy. Our method effectively corrects this mixed drift, achieving superior accuracy. Finally, we address pure pruning via DCT. Since cluster size $N_j$ is undefined in pure pruning, PA is not directly applicable; yet, radial drift associated with the global retention ratio can still arise. Our radial correction, dependent only on the global ratio $r$, remains applicable and recovers +6.4% accuracy over Vanilla at $r = 0.75$, suggesting that the radial component of our framework can extend beyond merging to broader reduction settings.

### 4.3. Robustness in Multi-modal Tasks

Multi-modal tasks impose stricter requirements on feature fidelity than classification, as they rely on the precise alignment between visual and textual embedding spaces. Covariate shift in the visual encoder can disrupt this alignment.

**Image-Text Retrieval: Preserving Geometric Alignment.** Retrieval tasks fundamentally rely on the precise dot-product alignment between visual and textual embedding spaces. As discussed in Eq. 8, token merging induces uncontrolled energy inflation, fundamentally distorting this geometry and disrupting the calibrated scale expected by the text encoder. Table 1 quantifies this impact. While Vanilla PiToMe boosts BLIP throughput by **2×** (48s → 24s), it incurs a steep accuracy penalty. PA fails to mitigate this decay. As shown in Figure 4, PA performs comparably to the Vanilla baseline across most settings (e.g., ALBEF on MS COCO), indicating that correcting attention probabilities alone is insufficient when the feature magnitude remains distorted. In specific cases like BLIP on Flickr30k, PA even collapses significantly below Vanilla, suggesting that its heuristic correction may amplify attention noise in sensitive models. In contrast, our $\sqrt{r}$-scaling calibrates energy dynamics, translating statistical stability into superior Pareto efficiency. On ALBEF (Flickr30k, $r = 0.9$), our method recovers +8.2 points in $R_{sum}$ over Vanilla, retaining 95.1% of uncompressed performance while maintaining identical latency reductions. Furthermore, as compression

*Table 2.* VQA Performance at $r = 0.9$. We report the Average Normalized Score (Avg. Norm.), benchmarked against the global best (100). Our method consistently recovers performance across diverse reasoning tasks (GQA, MME, etc.).

| Method | Scaling | GQA (Acc) | MME (Score) | VizWiz (Acc) | AI2D (Acc) | POPE (Acc) | MMMU (Acc) | InfoVQA (ANLS) | Avg. Norm. |
|---|---|---|---|---|---|---|---|---|---|
| ToMe | Vanilla | 55.38 | 1293.56 | 56.82 | 53.24 | 84.74 | 35.70 | 24.53 | 95.79 |
| | PA | 52.61 | 1170.23 | 56.45 | 50.55 | 82.88 | 35.10 | 24.03 | 92.26 |
| | $\sqrt{r}$ **(Ours)** | **55.91** | **1298.23** | **57.28** | **54.08** | **85.60** | **36.70** | **25.24** | **97.22** |
| PiToMe | Vanilla | 56.73 | 1303.98 | 56.67 | 53.30 | 84.30 | 36.10 | 25.03 | 96.56 |
| | PA | 55.15 | 1241.08 | 56.36 | 52.04 | 83.47 | 36.10 | 23.67 | 94.23 |
| | $\sqrt{r}$ **(Ours)** | **56.85** | **1332.39** | **57.15** | **53.63** | **84.36** | **36.60** | **24.74** | **97.12** |
| ToFu | Vanilla | 59.44 | 1390.89 | 55.47 | 54.79 | 87.31 | 37.10 | 24.28 | 98.63 |
| | PA | 58.93 | 1401.45 | 55.13 | 54.99 | 87.23 | 37.20 | 24.35 | 98.65 |
| | $\sqrt{r}$ **(Ours)** | **59.51** | **1403.52** | **55.58** | **55.18** | **87.39** | **37.40** | **25.03** | **99.46** |

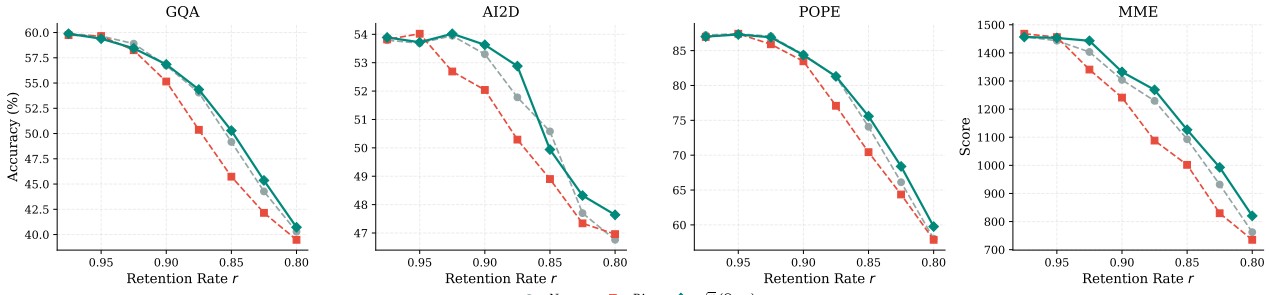

*Figure 5.* VQA Performance vs Compression Ratio. We evaluate PiToMe equipped with different scaling strategies across four diverse VQA benchmarks. Our $\sqrt{r}$-scaling (green) demonstrates superior robustness, effectively preserving the fine-grained semantic details necessary for complex reasoning tasks even at lower retention ratios.

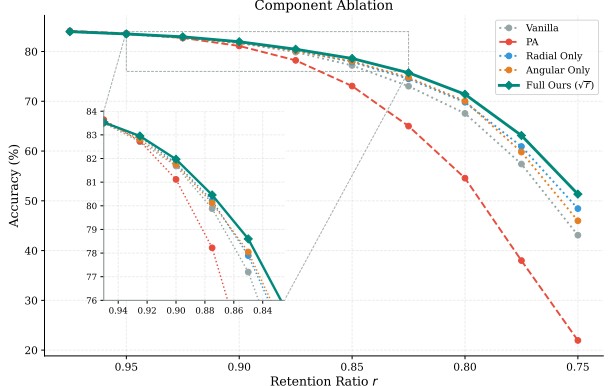

*Figure 6.* Component-wise Ablation. The angular-only variant outperforms PA by mitigating estimation bias, while the radial-only variant addresses energy drift. Their combination supports the independence of the two failure modes and the benefit of joint correction.

becomes aggressive ($r = 0.75$), our approach maintains a robust lead—achieving up to +166.4 higher $R_{sum}$ than PA on BLIP—indicating that statistical calibration is important for robust cross-modal alignment under aggressive compression.

**Visual Question Answering: Maintaining Semantic Detail.** Generative tasks like VQA require the model to retain fine-grained visual cues to support complex reasoning. Table 2 shows that at mild compression ($r = 0.9$), our $\sqrt{r}$-scaling consistently yields the highest Average Normalized Scores across diverse reduction algorithms. It is worth noting that the performance gaps at this stage are narrower compared to ImageNet. We attribute this to the inherent robustness of the 7B-parameter LLM (LLaVA-1.5), whose strong semantic priors can partially compensate for minor signal degradation. However, the instability of heuristics becomes evident as compression intensifies. Figure 5 reveals distinct degradation trajectories across benchmarks. Crucially, PA exhibits detrimental instability, frequently crossing below the uncorrected Vanilla baseline as $r$ decreases (e.g., in GQA, AI2D, and MME). This inversion suggests that PA's heuristic manipulation of attention logits introduces distribution shifts that misalign the MLLM projector, leading to performance degradation compared to the uncorrected baseline. In contrast, our method (green line) maintains statistical stability, yielding the flattest decay curve. This advantage is most pronounced on the POPE benchmark (measuring object hallucination), where our method outperforms PA (85.60 vs 82.88 on ToMe), suggesting that preserving the statistical fidelity of the residual stream helps reduce hallucinations and sustain precise visual reasoning.

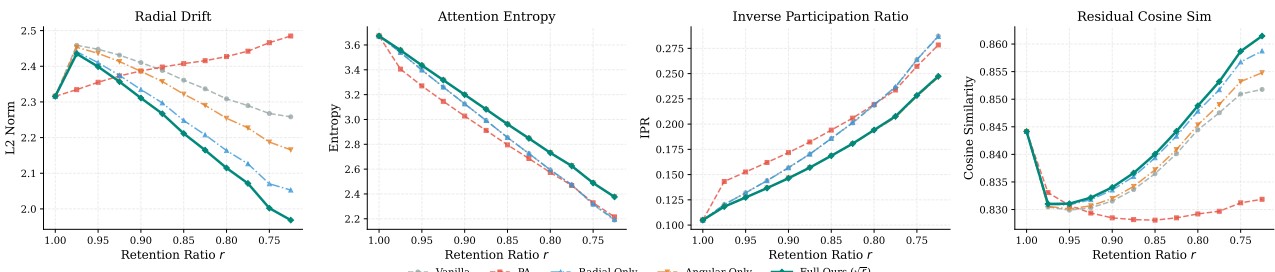

*Figure 7.* Statistical Consistency Analysis. **(1) Radial Drift (L2 Norm):** PA and angular-only exhibit energy inflation, while radial-only constrains the energy trajectory. **(2-3) Entropy & IPR:** The radial-only curve closely overlaps with Vanilla, indicating that radial scaling has little effect on the attention distribution. Meanwhile, angular-only restores part of the diversity reduced by PA. **(4) Residual Cosine Similarity:** The Full Method combines these benefits and achieves the highest alignment with the uncompressed model.

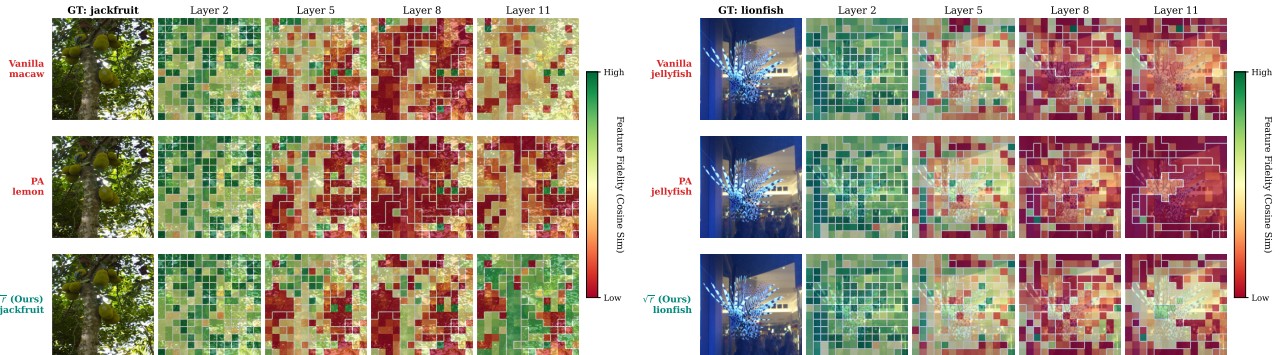

*Figure 8.* Visualizing Feature Fidelity. We map the cosine similarity between compressed and original tokens. **Green** indicates high fidelity (preserved semantics), **Red** indicates angular drift. Baselines (Vanilla, PA) exhibit severe drift in deep layers, leading to misclassifications (e.g., *Jackfruit → Macaw*). Our $\sqrt{r}$-scaling maintains high fidelity, preserving semantic integrity for correct classification.

## 4.4. Ablation Study and Mechanism Analysis

To validate the *Asymptotic Radial-Angular Decomposition* (Lemma 3.1), we decouple the radial ($\alpha = \sqrt{r}$) and angular ($\mathcal{T}$) corrections, analyzing their isolated impact on internal statistics (Figure 7) and performance (Figure 6).

**Distributional Restoration (Entropy & IPR).** We first examine the separation between radial and angular effects through the angular dynamics in Figure 7 (middle). The *radial-only* curve closely overlaps with the *vanilla* baseline, indicating that radial scaling alters magnitude while largely preserving the attention distribution. In contrast, PA causes a clear entropy drop. Our Full Method improves over the Vanilla baseline, restoring attention diversity (higher entropy, lower IPR) typically lost during merging. This supports that the LMMSE correction helps repair distributional damage rather than merely preserving the original merged state.

**Geometric Alignment (L2 Norm & Cosine).** The L2 norm (Figure 7, left) reveals the tension between inflation and collapse. Vanilla displays a characteristic hump followed by signal deflation, consistent with destructive interference under heterogeneous merging. PA exhibits monotonic upward drift. A nuanced trade-off appears at mild compression

($r > 0.9$): PA shows slight geometric superiority over our method. This can be explained by PA's passivity (identity mapping for unmerged tokens), whereas our asymptotic $\sqrt{r}$-scaling may induce a minor statistical over-correction. However, as $r$ decreases, PA's lack of radial correction leads to stronger drift, while our method acts as a statistical anchor; the observed deflation serves as a benign damping mechanism, reducing error amplification and improving signal stability. See Appendix C for details.

**Performance Synthesis.** Finally, Figure 6 provides additional evidence for the complementarity of the two corrections. The *angular-only* model outperforms PA, supporting the effectiveness of the LMMSE estimator. More importantly, the *full method* yields the strongest overall performance. The fact that our accuracy at $r = 0.95$ matches PA suggests that the minor geometric over-correction at high retention is benign, whereas the improved distributional preservation becomes critical for preventing sharp degradation at lower ratios.

## 4.5. Qualitative Analysis: Visualizing Feature Fidelity

To provide tangible intuition for the improvements, we visualize the layer-wise feature fidelity in Figure 8. The heatmaps reveal a characteristic U-shaped evolution consis-

tent with representational analyses of ViTs (Raghu et al., 2021; Caron et al., 2021): fidelity naturally drops in middle layers (2-8) as features undergo abstraction, but often partially recovers in deeper layers (8-11) as the network consolidates global semantics for classification.

Our $\sqrt{r}$-scaling consistently maintains the highest fidelity atop this trend. Crucially, we observe two distinct deficiencies in baselines. First, Vanilla consistently retains better structural integrity than PA, corroborating our quantitative findings that PA's zeroth-order correction introduces estimation bias. Second, PA exhibits a collapse in the final layers, deviating from the natural recovery trend and descending into severe angular drift.

Beyond angular fidelity, our method demonstrates superior spatial semantic alignment. As vividly shown in the *Lionfish* example (Layer 11), our merged super-tokens (visualized by grid borders) tightly encapsulate the semantic subject, effectively segmenting the object from the background. In contrast, PA produces fragmented clusters that bleed into the background. This suggests that statistical correction not only preserves feature direction but also encourages coherent grouping consistent with the object's topology.

## 5. Limitations and Operating Regimes

Our analysis characterizes $\sqrt{r}$-scaling as a training-free statistical calibration under standard Transformer operating conditions. The derivation is based on a leading-order reconstruction risk with finite second-moment and approximate variance-stationarity assumptions, which are supported in practice by normalization layers such as LayerNorm or RMSNorm. Architectures with explicit QK or logit normalization may further constrain attention-logit variance, potentially reducing the marginal benefit of the angular correction. Thus, the resulting correction should be interpreted as asymptotically optimal within this regime rather than as an unconditional optimum for arbitrary architectures. At extremely low retention ratios, irreversible information loss from highly heterogeneous clusters may dominate, although our scaling remains effective in mitigating the associated statistical drift. Finally, while the radial correction naturally applies to pruning and hybrid reduction through the global retention ratio, the angular correction relies on cluster-size information and is specific to merging-based settings. Extending this statistical reconstruction perspective to generation models with merge–unmerge pipelines, such as diffusion Transformers, remains an interesting direction for future work.

## 6. Conclusion

This work identifies radial drift and angular misalignment as two important sources of statistical shift in token reduction.

We introduce an asymptotic radial-angular decomposition to analyze these effects and derive a simple $\sqrt{r}$-scaling rule for value calibration and LMMSE-based logit shrinkage. The resulting correction is parameter-free, lightweight, and compatible with diverse reduction mechanisms, including merging, pruning, and hybrid fusion. Empirically, it consistently improves accuracy, robustness, and feature fidelity across vision and vision-language Transformer models under compressed inference.

## Impact Statement

This paper aims to advance efficient machine learning. By enabling high-fidelity token reduction, our work contributes to environmental sustainability (Green AI) by lowering the energy footprint of large-scale model inference. Furthermore, by mitigating hallucinations in compressed models, it enhances system reliability and promotes the accessibility of advanced AI on resource-constrained devices. We do not foresee specific negative societal consequences beyond general dual-use risks.

ACKNOWLEDGMENTS

This work was supported by the National Natural Science Foundation of China under Grant 62471394, 62306241 and 62576284.

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

# A. Detailed Theoretical Proofs

## A.1. Assumptions

Our theoretical analysis relies on two standard working assumptions that characterize the operating regime of stable Transformer representations.

**Assumption A.1** (Finite Second Moments). The projected key and value vectors $(\mathbf{k}_i, \mathbf{v}_i) \in \mathbb{R}^d$ have bounded second moments:

$$\mathbb{E}[\|\mathbf{v}_i\|^2] = C_v^2 < \infty, \quad \mathbb{E}[\|\mathbf{k}_i\|^2] = C_k^2 < \infty. \tag{12}$$

For simplicity in covariance calculations, we center the variables; nonzero means can be absorbed into the corresponding first-order terms without changing the variance-scaling argument.

*Justification.* This is a mild stability condition required for well-defined attention statistics. Standard initialization schemes, normalization layers, and stable training dynamics all encourage bounded activation moments in practical Transformer models.

**Assumption A.2** (Approximate Variance Stationarity via Normalization). The architecture employs a normalization mechanism, such as LayerNorm, RMSNorm, or an equivalent stabilizer, such that token representations entering key and value projections exhibit approximately stationary second-order statistics before and after merging:

$$\mathbb{E}[\|\text{Norm}(\mathbf{x}_i)\|^2] \approx \mathbb{E}[\|\text{Norm}(\mathbf{x}'_j)\|^2] \approx C^2, \tag{13}$$

where $\mathbf{x}'_j$ is the averaged representation of cluster $\mathcal{C}_j$.

*Justification.* This assumption formalizes the operating regime encouraged by normalization and residual architectures, rather than imposing an exact invariance condition. Prior analyses of deep signal propagation emphasize that stable deep networks require activation moments to remain controlled across layers (Schoenholz et al., 2017; Dong et al., 2021). In practice, normalization layers act as structural anchors that reduce the accumulation of variance drift after token reduction. Our $\sqrt{r}$ correction is derived within this approximately stationary regime, and Appendix D provides empirical diagnostics supporting this interpretation.

## A.2. Proof of Radial-Angular Decomposition

**Proposition.** *In high-dimensional spaces, the Mean Squared Error (MSE) between two vectors $\mathbf{O}$ and $\mathbf{O}'$ asymptotically decomposes into a radial (magnitude) error and an angular (distributional) error due to the concentration of measure.*

*Proof.* Let the original and merged output vectors be expressed in polar form: $\mathbf{O} = R\mathbf{u}$ and $\mathbf{O}' = R'\mathbf{u}'$, where $R, R' \geq 0$ are the magnitudes and $\mathbf{u}, \mathbf{u}'$ are unit direction vectors on the hypersphere $\mathbb{S}^{d-1}$. The squared Euclidean distance is:

$$\|\mathbf{O}' - \mathbf{O}\|^2 = \|\mathbf{O}'\|^2 + \|\mathbf{O}\|^2 - 2\langle\mathbf{O}', \mathbf{O}\rangle \tag{14}$$

$$= (R')^2 + R^2 - 2RR'\cos\theta, \tag{15}$$

where $\cos\theta = \langle\mathbf{u}', \mathbf{u}\rangle$. We utilize the algebraic identity $2RR' = (R')^2 + R^2 - (R' - R)^2$ to rewrite the cross term:

$$\|\mathbf{O}' - \mathbf{O}\|^2 = (R')^2 + R^2 - \left[(R')^2 + R^2 - (R' - R)^2\right]\cos\theta \tag{16}$$

$$= (R' - R)^2 \cos\theta + (R'^2 + R^2)(1 - \cos\theta). \tag{17}$$

Taking the expectation $\mathbb{E}[\cdot]$ over the data distribution:

$$\mathcal{E} = \mathbb{E}\left[(R' - R)^2 \cos\theta\right] + \mathbb{E}\left[(R'^2 + R^2)(1 - \cos\theta)\right]. \tag{18}$$

Under the high-dimensional concentration and weak radial-angular dependence assumptions, magnitude fluctuations are approximately decoupled from directional alignment. Thus, we approximate the expectation of the product as the product of expectations:

$$\mathcal{E} \approx \mathbb{E}[(R' - R)^2] \cdot \mathbb{E}[\cos\theta] + \mathbb{E}[R'^2 + R^2] \cdot \mathbb{E}[1 - \cos\theta]. \tag{19}$$

For a well-aligned approximation (where $\mathbf{O}'$ aims to reconstruct $\mathbf{O}$), the angle $\theta$ is small, so $\cos\theta \approx 1$. The first term simplifies to the pure **Radial Error**: $\mathbb{E}[(R' - R)^2]$. For the second term, we approximate $\mathbb{E}[R'^2 + R^2] \approx 2\mathbb{E}[R^2]$ (assuming

the radial correction aligns the scales). This yields the **Angular Error**: $2\mathbb{E}[R^2]\mathbb{E}[1 - \cos\theta]$. Combining these, we obtain the decomposition used in the main text:

$$\mathcal{E} \approx \underbrace{\mathbb{E}\left[(\|\mathbf{O}'\| - \|\mathbf{O}\|)^2\right]}_{\mathcal{E}_{\text{radial}}} + \underbrace{2\mathbb{E}[\|\mathbf{O}\|^2] \cdot \mathbb{E}\left[1 - \cos(\mathbf{O}', \mathbf{O})\right]}_{\mathcal{E}_{\text{angular}}}. \tag{20}$$

$\square$

### A.3. Proof of Radial Scaling (IPR Inflation)

**Proposition.** *Merging $N$ tokens into $M = rN$ clusters inflates the expected energy of the attention output by a factor bounded by $1/r$. We derive the scaling $\alpha = \sqrt{r}$ to counteract this upper-bound inflation.*

*Proof.* Let the attention output be $\mathbf{O} = \sum_{i=1}^{N} p_i \mathbf{v}_i$. Under Assumption 1 (Finite Second Moment), value vectors have zero mean and covariance $\boldsymbol{\Sigma}$ with $C^2 = \text{Tr}(\boldsymbol{\Sigma})$. The expected energy is:

$$\mathbb{E}[\|\mathbf{O}\|^2] = \sum_i \mathbb{E}[p_i^2]\text{Tr}(\boldsymbol{\Sigma}) + \sum_{i \neq j} \mathbb{E}[p_i p_j]\mathbb{E}[\mathbf{v}_i^\top \mathbf{v}_j]. \tag{21}$$

In high-dimensional spaces ($d \gg 1$), distinct vectors are asymptotically orthogonal due to the concentration of measure. We thus simplify the cross-terms $\mathbb{E}[\mathbf{v}_i^\top \mathbf{v}_j] \approx 0$. The energy becomes proportional to the *Inverse Participation Ratio (IPR)* of the attention weights:

$$\mathcal{E}_{orig} \approx C^2 \cdot \mathbb{E}\left[\sum_{i=1}^{N} p_i^2\right]. \tag{22}$$

Now consider the merged output $\mathbf{O}'$. Let the cluster $\mathcal{C}_j$ have an average size $k \approx 1/r$. The merged probability is $P_j = \sum_{m \in \mathcal{C}_j} p_m$, and the projected values of merged tokens have approximately stationary second moments after normalization (Assumption 2). The new energy is governed by the squared sum of probabilities:

$$\mathcal{E}_{new} = C^2 \cdot \mathbb{E}\left[\sum_{j=1}^{M} P_j^2\right] = C^2 \cdot \mathbb{E}\left[\sum_{j=1}^{M} \left(\sum_{m \in \mathcal{C}_j} p_m\right)^2\right]. \tag{23}$$

We analyze the inflation using the Cauchy-Schwarz inequality. For any cluster $\mathcal{C}_j$:

$$P_j^2 = \left(\sum_{m \in \mathcal{C}_j} p_m \cdot 1\right)^2 \leq k \sum_{m \in \mathcal{C}_j} p_m^2. \tag{24}$$

The equality holds if and only if attention weights are homogeneous ($p_m \approx p$), which is the design objective of token merging (grouping similar tokens). In this high correlation regime, the inflation is maximized:

$$\mathcal{E}_{new} \approx \sum_{j=1}^{M} k \sum_{m \in \mathcal{C}_j} p_m^2 = k \sum_{i=1}^{N} p_i^2 = \frac{1}{r}\mathcal{E}_{orig}. \tag{25}$$

As compression becomes aggressive ($r \to 0$) and clusters become heterogeneous, the strict inequality holds ($\mathcal{E}_{new} < \frac{1}{r}\mathcal{E}_{orig}$). To ensure stability, we calibrate against the maximum theoretical inflation. Applying a scaling factor $\alpha$ to $\mathbf{v}'$:

$$\alpha^2 \cdot \frac{1}{r} = 1 \implies \alpha = \sqrt{r}. \tag{26}$$

This scaling bounds the upper-limit inflation term, mitigating runaway energy amplification while allowing benign deflation in heterogeneous regimes. $\square$

**Remark: Reconciliation with Empirical Deflation.** It is crucial to distinguish between the *theoretical inflation pressure* derived above and the *observed trajectory* of baselines. The derived factor $1/r$ represents the combinatorial inflation caused by aggregating probability mass. However, in practice, merging heterogeneous tokens introduces a counter-acting force: destructive interference. When distinct vectors $v_i, v_j$ are averaged, their magnitude decays: $||\frac{v_i+v_j}{2}||^2 < \frac{||v_i||^2+||v_j||^2}{2}$.

- **Vanilla Dynamics:** In Vanilla ToMe, these two forces compete. At high $r$, homogeneity minimizes interference, allowing inflation to dominate (creating the "hump" in Fig. 7). At low $r$, heterogeneity maximizes interference, overpowering inflation and causing the energy to collapse naturally.

- **The Role of $\sqrt{r}$:** Our scaling $\alpha = \sqrt{r}$ is designed to neutralize the upper-bound combinatorial inflation component. By removing this artificial upward pressure, the scaling allows the model to follow a more conservative energy trajectory under heterogeneous merging. Thus, we do not aim to force the norm to match the uncompressed case exactly, but rather to reduce aggregation-induced amplification while preserving the remaining signal structure.

### A.4. Proof of Angular Scaling (LMMSE Estimator)

**Proposition.** *Under a standardized exchangeable model for intra-cluster logits, the reliability of a merged cluster logit as a proxy for token-level logits scales as $\rho \approx \sqrt{r}$.*

*Proof.* We analyze centered logits relative to the cluster-size prior. Let

$$\xi_i = z_i - \ln N_j, \qquad \mathbb{E}[\xi_i] = 0, \qquad \mathrm{Var}(\xi_i) = \sigma^2, \tag{27}$$

where $\xi_i$ captures token-level logit variation within cluster $\mathcal{C}_j$. For a cluster of average size $k \approx 1/r$, the raw averaged residual is

$$u_j = \frac{1}{k} \sum_{i=1}^{k} \xi_i. \tag{28}$$

Under weak intra-cluster residual correlations,

$$\mathrm{Var}(u_j) \approx \frac{\sigma^2}{k}, \qquad \mathrm{Cov}(u_j, \xi_i) \approx \frac{\sigma^2}{k}. \tag{29}$$

In pre-normalized Transformer blocks, the merged representation is normalized before producing attention logits. Therefore, the cluster-level logit entering attention is better modeled as a standardized observation with comparable variance to token-level logits. Denote this standardized observation by

$$\tilde{\xi}_j = \sqrt{k}\, u_j, \qquad \mathrm{Var}(\tilde{\xi}_j) \approx \sigma^2. \tag{30}$$

Its correlation with an individual token-level residual is then

$$\rho = \mathrm{Corr}(\tilde{\xi}_j, \xi_i) \approx \frac{1}{\sqrt{k}} = \sqrt{r}. \tag{31}$$

For standardized variables, the LMMSE coefficient equals the correlation coefficient. Thus, the merged logit should be trusted with reliability weight $\rho \approx \sqrt{r}$ when used as a proxy for token-level logits. Identifying the standardized residual observation with the normalized merged-logit residual, i.e., $\bar{z}_j - \ln N_j \approx \tilde{\xi}_j$, the LMMSE estimate is obtained by shrinking $\bar{z}_j$ toward the cluster-size prior:

$$\hat{z}_j = \rho \bar{z}_j + (1 - \rho) \ln N_j. \tag{32}$$

Substituting $\rho = \sqrt{r}$ yields

$$\hat{z}_j = \sqrt{r}\, \bar{z}_j + (1 - \sqrt{r}) \ln N_j. \tag{33}$$

This derivation should be interpreted as a leading-order reliability model. If a cluster contains a strong shared signal component, the true reliability can lie between $\sqrt{r}$ and 1; using $\sqrt{r}$ gives a conservative shrinkage rule under heterogeneous merging. $\qquad \square$

## A.5. Analysis of Proportional Attention Bias

**Corollary.** *Proportional Attention (PA) incurs a second-order estimation bias approximately proportional to the intra-cluster logit variance.*

*Proof.* Let the true log-probability mass of cluster $\mathcal{C}_j$ be

$$y = \ln \left( \sum_{i=1}^{N_j} e^{z_i} \right). \tag{34}$$

We perform a second-order Taylor expansion around the cluster mean $\bar{z}_j$. Let

$$z_i = \bar{z}_j + \delta_i, \quad \sum_{i=1}^{N_j} \delta_i = 0, \quad \sigma_j^2 = \frac{1}{N_j} \sum_{i=1}^{N_j} \delta_i^2. \tag{35}$$

Then

$$y = \ln \left( \sum_{i=1}^{N_j} e^{\bar{z}_j + \delta_i} \right) = \bar{z}_j + \ln \left( \sum_{i=1}^{N_j} e^{\delta_i} \right) \tag{36}$$

$$\approx \bar{z}_j + \ln \left( \sum_{i=1}^{N_j} \left( 1 + \delta_i + \frac{1}{2} \delta_i^2 \right) \right) \tag{37}$$

$$= \bar{z}_j + \ln \left( N_j + \frac{1}{2} \sum_{i=1}^{N_j} \delta_i^2 \right) \tag{38}$$

$$= \bar{z}_j + \ln N_j + \ln \left( 1 + \frac{1}{2} \sigma_j^2 \right). \tag{39}$$

Using $\ln(1 + x) \approx x$ for small variance terms gives

$$y \approx (\bar{z}_j + \ln N_j) + \frac{1}{2} \sigma_j^2. \tag{40}$$

The PA estimator is $\hat{y}_{\text{PA}} = \bar{z}_j + \ln N_j$, so the second-order residual is

$$y - \hat{y}_{\text{PA}} \approx \frac{1}{2} \sigma_j^2. \tag{41}$$

As compression becomes more aggressive, clusters tend to contain more heterogeneous tokens, increasing $\sigma_j^2$ and making this approximation error more pronounced. $\qquad\square$

## A.6. Resolution of Overconfidence Bias via LMMSE

**Proposition.** *The LMMSE shrinkage factor $\sqrt{r}$ mitigates the overconfidence induced by trusting noisy merged logits too strongly.*

*Proof.* Appendix A.5 shows that the PA estimator omits a second-order variance term proportional to $\sigma_j^2$. This residual indicates that PA does not model the uncertainty induced by intra-cluster logit variance. Empirically, this can lead to sharper and less stable attention distributions under heterogeneous merging.

Ignoring the additive cluster-size prior for clarity, our angular correction applies a shrinkage to the averaged cluster logits:

$$\tilde{p}_j = \text{Softmax}(\sqrt{r}\,\bar{z}_j)_j. \tag{42}$$

Equivalently, this can be viewed as applying a temperature

$$T = \frac{1}{\sqrt{r}} > 1 \tag{43}$$

---

**Algorithm 1** Self-Attention with $\sqrt{r}$-Scaling

---

1: **Input:** Input Tokens $X \in \mathbb{R}^{N \times d}$, Retention Ratio $r \in (0, 1]$
2: **Output:** Reduced Output $O \in \mathbb{R}^{M \times d}$
3: **Step 1: Reduction & Projection**
4: Indices, $S \leftarrow \text{ComputeReductionPolicy}(X, r)$                                      {Determine indices from input features}
5: $M \leftarrow \text{size}(S)$; $r_{curr} \leftarrow M/N$                                     {Calculate effective retention ratio}
6: $X' \leftarrow \text{Merge}(X, \text{Indices})$                                        {Reduce input tokens $(M \times d)$}
7: $Q', K', V' \leftarrow \text{Linear}(X')$                                          {Apply linear projections}
8: **Step 2: Statistical Calibration (Ours)**
9: $V' \leftarrow V' \cdot \sqrt{r_{curr}}$                                              {**2a. Radial:** Calibrate value energy}
10: $A \leftarrow (Q' \cdot (K')^{\top})/\sqrt{d}$                                  {Compute raw attention logits $(M \times M)$}
11: $A \leftarrow \sqrt{r_{curr}} \cdot A + (1 - \sqrt{r_{curr}}) \cdot \ln(S)^{\top}$      {**2b. Angular:** LMMSE correction w/ size prior}
12: **Step 3: Output Computation**
13: $O \leftarrow \text{Softmax}(A) \cdot V'$                                      {Standard attention output}
14: **return** $O$

---

to the cluster-level logits. A larger temperature smooths the softmax distribution, reducing the dominance of overconfident logits and increasing entropy. Thus, the LMMSE shrinkage acts as a temperature-like calibration that counteracts the sharpening effect caused by noisy heterogeneous clusters.

In short, PA fully trusts the averaged logit, whereas our estimator assigns it a reliability weight $\sqrt{r} < 1$ under compression. This provides a principled smoothing mechanism consistent with the second-order bias identified in Eq. 41.        □

## B. Implementation Details and Algorithm

In this section, we provide the detailed algorithmic implementation of our proposed $\sqrt{r}$-scaling framework. Our method is designed to be a plug-and-play statistical calibration module that integrates seamlessly with existing token reduction mechanisms (e.g., ToMe, ToFu, PiToMe, and etc.).

### B.1. Algorithmic Overview

Algorithm 1 outlines the modified Self-Attention forward pass equipped with our statistical corrections. The core distinction from standard Token Merging lies in two lightweight scalar operations:

- **Radial Correction (Line 9):** We scale the merged value vectors $V'$ by $\alpha = \sqrt{r}$ to counteract the aggregation-induced energy inflation (counteracting the $1/r$ drift derived in Eq. 7).

- **Angular Correction (Line 12):** We calibrate the attention logits $A$ using the LMMSE estimator $\mathcal{T}(z) = \sqrt{r}z + (1 - \sqrt{r}) \ln N_j$. This shrinks the noisy logits towards the geometric prior (cluster size $N_j$) to restore distributional fidelity.

Crucially, unlike heuristic methods that may require complex auxiliary losses or iterative optimization, our corrections are closed-form and incur negligible computational overhead, preserving the efficiency benefits of token merging.

### B.2. Complexity Analysis

The computational cost of our correction is strictly marginal. For **Radial Correction**, the scaling is a broadcasted element-wise multiplication on $V' \in \mathbb{R}^{M \times d}$, requiring $M \times d$ FLOPs. For **Angular Correction**, the operation involves scaling the attention matrix $A \in \mathbb{R}^{M \times M}$ and adding the bias vector $\ln S$, requiring approximately $2M^2$ FLOPs.

To contextualize this overhead, we consider the standard self-attention mechanism, which is dominated by two matrix multiplications ($QK^{\top}$ and $AV$), incurring a total cost of approximately $4M^2 d$ FLOPs. Consequently, the complexity ratio of our correction to the attention mechanism scales as:

$$\frac{\text{Cost}_{\text{Ours}}}{\text{Cost}_{\text{Attn}}} \approx \frac{2M^2 + Md}{4M^2 d} \approx \frac{1}{2d}. \tag{44}$$

For a typical head dimension $d = 64$, this ratio represents an overhead of less than $0.8\%$. This theoretical efficiency is empirically verified in two ways:

- **GFLOPs Trajectories:** As shown in Figure 3 of the main text, the GFLOPs curves (dashed lines) for Vanilla, PA, and our method overlap, confirming zero structural overhead.

- **Wall-Clock Latency:** Table 1 demonstrates that our method achieves identical inference times to the Vanilla baseline (e.g., matching the 24s throughput of BLIP at $r = 0.9$ and roughly 1m37s for ALBEF), validating that the scalar corrections incur no measurable latency penalty in practice.

## C. Extended Analysis of Radial Energy Dynamics

In the main text (Figure 7, left), we observed distinct energy trajectories across different scaling strategies. In this section, we provide a more detailed analysis of these trends to reconcile the theoretical $1/r$ inflation pressure with the empirical observations of deflation in certain regimes, thereby clarifying the role of the proposed $\sqrt{r}$-scaling.

**1. Evidence of Inflationary Pressure: The Case of Proportional Attention.** The energy inflation mechanism derived in Eq. 7 is supported by the trajectory of Proportional Attention (PA, red dashed line). While PA introduces a bias to correct attention logits, it applies no correction to the value vectors ($\alpha = 1$). Consequently, it remains exposed to radial drift. As illustrated in Figure 7, PA exhibits a monotonic increase in L2 norm as retention decreases (rising from $\sim 2.3$ to $> 2.5$). Notably, this inflation exceeds that of the Vanilla baseline. This suggests that PA's angular correction, by sharpening the attention distribution (reducing entropy), can inadvertently amplify the combinatorial inflation of the attention weights. This observation supports the need for explicit radial calibration, since the aggregation of probability mass into fewer clusters can dominate the variance reduction inherent in vector averaging and lead to energy amplification.

**2. The Hump and Deflation of Vanilla Merging.** The Vanilla baseline (grey dashed line), representing uncorrected token merging, reveals the stochastic tension between aggregation-induced inflation and interference-induced deflation.

- **The Inflationary Hump ($1.0 > r > 0.8$):** In the high-similarity regime, the L2 norm initially rises. This hump indicates that when merged tokens are relatively homogeneous, the $1/r$ inflation pressure can outweigh the variance reduction from averaging, providing empirical evidence for radial drift.

- **Destructive Deflation ($r < 0.8$):** As merging becomes more aggressive, the system is forced to combine heterogeneous tokens. The destructive interference between dissimilar vectors causes the variance reduction to accelerate, eventually overpowering the inflation pressure. The curve subsequently decreases.

- **Performance Implication:** At aggressive compression ratios ($r = 0.75$), Vanilla outperforms PA (see Figure 2). This suggests that moderate deflation can be less harmful to deep signal propagation than the stronger energy amplification exhibited by PA.

**3. $\sqrt{r}$-Scaling as a Statistical Damper.** Our method (green solid line) applies a closed-form scaling of $\alpha = \sqrt{r}$. While derived to counteract the upper-bound inflation under ideal homogeneity, empirical results show a monotonic decrease in energy under heterogeneous merging. In light of the comparative analysis above, this trajectory can be interpreted as a conservative statistical damping mechanism rather than a failure of calibration:

- **Damping vs. Uncontrolled Dynamics:** Unlike Vanilla, whose deflation is a passive consequence of signal interference, our method explicitly removes the combinatorial inflation pressure. This helps counteract the inflation tendency that drives PA's degradation.

- **Safe-Side Behavior:** The superiority of our method over both PA and Vanilla in the evaluated regimes suggests that conservative damping is beneficial for residual-stream stability. By accepting a benign reduction in magnitude, our approach mitigates noise amplification under heterogeneous clustering and preserves feature fidelity across the tested compression levels.

# D. Additional Diagnostic Analyses

We include additional diagnostics to clarify the operating regimes of our assumptions and corrections. These analyses are not used for tuning; the scaling factor remains exactly $\sqrt{r}$. Unless otherwise specified, diagnostics are measured on ViT-B with raw ToMe at $c = 16$ merged tokens per layer.

## D.1. Variance Stationarity before and after Normalization

To examine the role of normalization, we apply raw token merging without scaling and measure the feature variance ratio across ViT-B blocks before and after normalization. As shown in Table 3, pre-normalization features exhibit noticeable drift in deeper blocks, whereas post-normalization statistics remain substantially more stable. This supports our use of normalization as an approximate structural anchor, while also highlighting that stationarity is not an exact invariance.

*Table 3.* Variance ratio diagnostic across ViT-B blocks under raw token merging. Ratios are measured relative to the uncompressed model.

| Block Index | 0 | 4 | 6 | 8 | 10 | 11 |
|---|---|---|---|---|---|---|
| Pre-Norm Ratio | 1.00 | 0.96 | 1.01 | 1.30 | 1.45 | 1.32 |
| Post-Norm Ratio | 0.99 | 0.99 | 0.98 | 0.95 | 0.80 | 0.79 |

## D.2. Inflation–Deflation Transition under Cluster Heterogeneity

We further examine the raw post-merge energy ratio as a function of intra-cluster cosine similarity. Table 4 shows that highly homogeneous clusters tend to exhibit inflation, while low-similarity clusters are dominated by destructive interference and often deflate. This explains why the theoretical $1/r$ factor should be interpreted as an upper-bound inflation pressure rather than a universal empirical trajectory.

*Table 4.* Raw post-merge energy ratio under different intra-cluster similarity regimes.

| Cosine Similarity Range | $> 0.92$ | 0.88–0.92 | 0.83–0.88 | $< 0.83$ |
|---|---|---|---|---|
| Avg. Energy Ratio | 1.039 | 1.032 | 1.051 | 0.964 |
| Deflated Clusters ($< 1.0$) | 0.0% | 26.4% | 66.2% | 76.5% |
| Phase | Inflation | Marginal | Transition | Deflation |

## D.3. Models Trained with Token Merging

When a model is trained from scratch with a fixed token merging configuration, part of the distribution shift can be absorbed into the learned weights. To test whether explicit scaling remains useful beyond the training compression level, we evaluate official ToMe DeiT-S weights trained with ToMe+PA and vary the inference-time compression. Since the PA bias has already been absorbed during training, we apply only the radial correction on top of PA. Table 5 shows that the correction has little effect at the training setting but becomes beneficial under stronger zero-shot compression.

*Table 5.* Zero-shot higher-compression evaluation on ToMe DeiT-S trained with ToMe+PA.

| Inference Merged Tokens $c$ | 13 | 16 | 20 | 24 | 30 |
|---|---|---|---|---|---|
| Vanilla | 77.5 | 75.9 | 70.4 | 56.6 | 40.8 |
| PA | 79.4 | 79.1 | 77.3 | 72.2 | 61.0 |
| PA + Radial $\sqrt{r}$ | 79.3 | 79.1 | 77.4 | 72.7 | 62.3 |

