# OpenReview forum: "Statistically Calibrated Scaling for Token Merging in Transformers"
_ICML.cc/2026/Conference — ICML 2026 regular_

### Official Review · Reviewer_iQDt · 2026-03-04

**Soundness:** 3
**Presentation:** 2
**Significance:** 3
**Originality:** 3
**Overall Recommendation:** 4
**Confidence:** 3

**Summary:**

This paper investigates the severe performance degradation of Vision Transformers under aggressive token merging regimes, identifying covariate shift in the residual streams as the primary underlying cause. Authors decomposed this shift into two orthogonal components: radial and angular shift. Relying on assumptions of finite second moments and variance stationarity, this paper frames token merging as a statistical reconstruction problem and derives a closed-form scaling factor. This single scalar is applied to mathematically scale the merged value vectors and shrink the merged attention logits toward a cluster-size prior, theoretically aiming to restore energy conservation and distributional fidelity without the need for additional training or fine-tuning.

**Compliance With Llm Reviewing Policy:**

Affirmed.

**Final Justification:**

Most of my concerns have been solved; however, I think this paper still lacks outstanding insight. Thus, I maintain my original rating.

**Key Questions For Authors:**

- Q1. The theoretical derivation hinges on an assumption of variance stationarity under token merging, yet the covariance structure should in principle change substantially when heterogeneous tokens are merged. Could the authors provide empirical evidence or diagnostic analyses demonstrating that variance stationarity holds approximately in practice, especially at higher compression ratios?
- Q2. The text motivates the correction via a 1/r energy inflation effect, but the appendix suggests that real-world behavior often shows energy deflation due to destructive interference.
Could the authors precisely characterize the conditions (e.g., token similarity thresholds, compression ratios, layer types) under which inflation vs. deflation occurs?

**Limitations:**

yes

**Strengths And Weaknesses:**

**Strengths**
- The idea of reframing token merging as an estimation problem is interesting, and the radial–angular decomposition gives a clearer interpretation than heuristic rules like Proportional Attention. Treating magnitude and direction separately feels conceptually clean and helps explain why the method behaves more predictably.
- The fact that the correction is purely arithmetic and does not require any retraining is practically useful. This makes the technique easy to plug into existing pipelines without worrying about extra compute or fine-tuning.
- I appreciate the shift in perspective from choosing which tokens to merge toward calibrating the merged features themselves. This continuous correction viewpoint seems more scalable than discrete token selection and could inspire further work.

**Weakness**
- The theoretical development depends crucially on an assumption of variance stationarity, yet aggressive token merging substantially alters the covariance structure of the underlying feature distribution. This makes the assumption internally circular and weakly supported under realistic compression settings.
- The derivation of the 1/r energy inflation effect assumes that merged tokens remain highly homogeneous. In practice, large compression ratios inevitably combine semantically heterogeneous tokens, where destructive interference dominates and invalidates the inflation argument.
- The appeal to concentration-of-measure phenomena to justify near-orthogonality is difficult to justify for typical attention head dimensions (d=64 or 128), which are far below the asymptotic regimes required for such theoretical guarantees. As a result, the supporting argument becomes tenuous.
- In the broader landscape of model compression, post-hoc token merging is increasingly overshadowed by more competitive strategies such as structured pruning, low-bit quantization, or architectures designed for inherent efficiency. The manuscript provides limited discussion of how the proposed method fits into this evolving ecosystem.
- Although the method shows relative gains at extreme compression ratios, the absolute accuracy in these regimes degrades sharply. This raises concerns about the practical relevance of deploying models operating at such diminished performance levels.

---

> ### Author Rebuttal · Authors · 2026-03-26
>
> We deeply appreciate Reviewer iQDt for the rigorous theoretical scrutiny and for finding our feature-space perspective conceptually clean and practically useful. Below, we address your insightful feedback.
>
> **Q1 & W1: Variance Stationarity and Covariance Shift**
> We agree that merging heterogeneous tokens substantially alters the internal covariance structure. However, our Variance Stationarity assumption relies heavily on architectural constraints—specifically LayerNorm.
>
> To empirically validate this, we conducted a diagnostic stress test extracting feature variances across ViT-B layers before and after LayerNorm. To observe the unmitigated drift, we applied raw Token Merging (merging $c=16$ tokens per layer) *without* any scaling interventions:
>
> | Block Index | 0 | 4 | 6 | 8 | 10 | 11 |
> | :--- | :---: | :---: | :---: | :---: | :---: | :---: |
> | **Pre-Norm Ratio** (Shift) | 1.00 | 0.96 | 1.01 | 1.30 | 1.45 | 1.32 |
> | **Post-Norm Ratio** (Anchor)| 0.99 | 0.99 | 0.98 | 0.95 | 0.80 | 0.79 |
>
> *Empirical Conclusion:* As you insightfully predicted, the Pre-Norm variance drifts heavily (inflating up to 1.45x), providing direct evidence of compounding covariance shift. However, LayerNorm acts as a strict structural anchor. While deep Post-Norm layers drop to ~0.80 (because the learned affine parameter $\gamma$ projects the shifting semantic distribution of surviving tokens slightly differently), the normalization successfully breaks the exponential compounding drift. This block-wise structural resetting validates why our localized *intra-block* geometric correction is mathematically sound.
>
> **Q2 & W2: 1/r Inflation vs. Deflation (Active Damping Mechanism)**
> You astutely noted the tension between the theoretical $1/r$ inflation and the real-world deflation caused by destructive interference among heterogeneous tokens. To *precisely characterize* this boundary as requested, we extracted the raw Post-Merge Energy Ratio (again, without scaling interventions) against the Intra-Cluster Cosine Similarity:
>
> | Cosine Similarity Range | > 0.92 | 0.88 - 0.92 | 0.83 - 0.88 | < 0.83 |
> | :--- | :---: | :---: | :---: | :---: |
> | **Avg Energy Ratio** | 1.039 | 1.032 | 1.051 | 0.964 |
> | **Deflated Clusters** (<1.0)| 0.0% | 26.4% | 66.2% | 76.5% |
> | **Physical Phase** | Strict Inflation | Marginal | Transition Zone | Strict Deflation |
>
> *Why Our Method Works:* The data reveals a clear non-linear phase transition. Highly homogeneous clusters ($>0.92$) strictly inflate, aligning with the $1/r$ theory. Conversely, as similarity drops below $0.83$ (inevitable in deep layers under aggressive compression), destructive interference takes over, and 76.5% of clusters strictly deflate.
> This precise characterization proves why our $\sqrt{r}$ scaling is robust. Proportional Attention (PA) blindly assumes homogeneity and forces a $1/r$ multiplier, which catastrophically amplifies noise in the heterogeneous transition zones. In contrast, our $\sqrt{r}$ scalar acts as an **"Active Damping"** mechanism. It gracefully accommodates natural deflation, firmly preventing variance explosion while establishing a bounded energy floor.
>
> **W3: Concentration of Measure at d=64**
> While our *radial correction* operates on the full embedding dimension ($D=384/768$) where concentration is exceptionally strong, we acknowledge that our *angular correction*—derived at standard attention head dimensions ($d=64$)—is indeed a finite-dimensional approximation. However, empirically, the dot product distribution in standard Transformers remains sufficiently concentrated for our zero-mean cross-term approximation to hold reasonably well. This allows our LMMSE estimator to effectively shrink the angular shift and provide statistically significant stabilization.
>
> **W4: Broader Landscape (Pruning and Quantization)**
> Token merging dynamically reduces sequence length ($N$), which is fundamentally orthogonal to static weight reduction techniques like structured pruning or low-bit quantization. Our arithmetic correction can be seamlessly integrated atop quantized models to yield compounded inference acceleration without conflict. We will add a discussion expanding on how our method fits into this broader model compression ecosystem in the final manuscript.
>
>
> **W5: Degraded Absolute Accuracy at Extreme Compression**
> We agree that accuracy degrades under extreme compression, but this is a fundamental information-theoretic bottleneck inherent to all token reduction methods, not a unique limitation of ours. Our approach already outperforms existing baselines at moderate, practical compression rates, and this advantage is only further amplified under extreme conditions.
>
> ---
> **Conclusion**
> We are exceptionally grateful for your feedback. We will incorporate these nuanced diagnostics into the final manuscript and *hope they merit a higher score.* Thank you for your time and expertise.

---

> > ### Author Rebuttal · Reviewer_iQDt · 2026-04-01
> >
> > I maintain the original rating.

---

### Official Review · Reviewer_ebPh · 2026-03-11

**Soundness:** 4
**Presentation:** 3
**Significance:** 3
**Originality:** 3
**Overall Recommendation:** 5
**Confidence:** 4

**Summary:**

This paper studies the performance degradation that occurs when token merging is used to accelerate Transformer inference. Token merging reduces the number of tokens processed by self-attention, improving efficiency, but often causes significant accuracy drops under high compression. The authors argue that this degradation arises from statistical distortions introduced during token aggregation, specifically energy drift in value vectors and bias in attention logits. To address this, they reformulate token merging as a statistical reconstruction problem and show that the reconstruction error can be decomposed into radial (magnitude) and angular (distribution) components. Based on this analysis, they derive a simple correction consisting of scaling merged value vectors by √r and adjusting attention logits using a cluster-size prior, which stabilizes both feature magnitude and attention distributions. Experiments on Vision Transformers demonstrate that this statistically motivated correction significantly improves accuracy and feature fidelity compared to existing heuristic methods under aggressive token compression.

**Compliance With Llm Reviewing Policy:**

Affirmed.

**Key Questions For Authors:**

The approach appears likely to transfer to other Transformer variants that employ token reduction. However, the paper currently validates it only on Vision Transformers on simple classification and retrieval tasks. It would be valuable to see how the method performs on other architectures, such as LLMs, image/video gens model.

**Limitations:**

Yes, the author included 1 section for limitation

**Strengths And Weaknesses:**

**Strengths**
- Overall I like the motivation of the paper, particularly the goal of recovering distortions in the feature space after token compression which is mostly ignored by token merging methods.

- The proposed method introduces negligible additional computational overhead, requiring only simple scalar scaling and a logit transformation. This design preserves the key advantage of token merging—efficient inference—while improving model stability.

- The experimental results show notable accuracy improvements at high compression ratios, a regime where prior token merging approaches typically suffer from significant performance degradation.

**Weaknesses**

- I do not observe any major weaknesses in the current version of the paper.

---

> ### Author Rebuttal · Authors · 2026-03-25
>
> We sincerely thank Reviewer ebPh for the "Accept" recommendation and for recognizing the core motivation, effectiveness, and computational efficiency of our method. Regarding your insightful question on generalization:
>
> **1. Performance on LLMs:** We actually have validated our method on LLM-based architectures. As presented in **Section 4.3 (Table 2 and Figure 5)**, we evaluated our approach on **LLaVA** (a Large Multimodal Model built upon an LLM backbone) for Visual Question Answering (VQA) tasks. The results demonstrate that our theoretically derived scaling successfully transfers to LLM-driven architectures, significantly mitigating accuracy degradation under aggressive compression.
>
> **2. Transferability to Image/Video Generation Models (e.g., DiTs):**
> This is a highly forward-looking question. Generative models (like DiTs) typically require the token sequence length to remain strictly constant at the output to preserve spatial resolution. Consequently, token merging in DiTs often operates as a "merge before attention, unmerge after" pipeline, which structurally differs from the permanent reduction paradigm analyzed in our current framework.
>
> However, the fundamental perspective proposed in our paper is highly relevant to DiTs. Prior token merging research has predominantly focused on *token selection and matching strategies*. Our work highlights a critical blind spot: the necessity of analyzing and correcting the resulting **feature space distortions**. Since generative models are notoriously sensitive to activation magnitudes, we believe our statistical reconstruction perspective offers a promising theoretical lens for exploring stable token merging schemes in future DiT research.
>
> ---
> **Conclusion**
> We sincerely thank you again for championing our paper and for recognizing the fundamental shift our work introduces—moving the focus of token merging from heuristic *token selection* to *feature space reconstruction*. Given your strong endorsement of our method's novelty, efficiency, and broad impact, *if you find it appropriate, we kindly hope you might consider raising your score to further champion our work.* Thank you for your invaluable feedback.

---

> > ### Author Rebuttal · Reviewer_ebPh · 2026-04-04
> >
> > All my concerns are well addressed, and I will keep my score.

---

### Official Review · Reviewer_Rqbm · 2026-03-13

**Soundness:** 3
**Presentation:** 3
**Significance:** 3
**Originality:** 2
**Overall Recommendation:** 4
**Confidence:** 3

**Summary:**

The authors propose an improvement over existing token merging schemes. By framing token merging as a statistical reconstruction problem, the authors solve for the optimal logit bias and value scaling parameters. Extensive evaluation on Image Classification on ImageNet-1K using ViT, Image-Text Retrieval on MS-COCO and Flickr30k, and Visual Question Answering across a wide range of "base" token merging schemes (DCT, DiffRate, PiToMe, ToMe and ToFu) demonstrate the effectiveness of the proposed method.

**Compliance With Llm Reviewing Policy:**

Affirmed.

**Final Justification:**

Initial concerns around presentation are resolved granted that the changes proposed by the authors in the rebuttal are reflected in the final revision.

**Key Questions For Authors:**

Q1: Given the many assumptions and approximations used in the results (e.g., d>>1), I would imagine that there are hidden constant factors. In your experiments, is sqrt(0.75) the actual scaling parameter used?

**Limitations:**

Yes

**Strengths And Weaknesses:**

S1: Proofs and experimental setup make sense to me.
W1: In the first paragraph of Section 1, could you clarify what "residual stream" means?
S2: The proposed method is a meaningful improvement over vanilla token merging.

---

> ### Author Rebuttal · Authors · 2026-03-25
>
> We sincerely thank Reviewer Rqbm for the positive evaluation, the "Weak Accept" rating, and for acknowledging the comprehensive nature of our experiments and the logic of our proofs. We are glad you find our method to be a meaningful improvement. Below, we address your questions directly.
>
> **W1: Clarification of "Residual Stream"**
> In Transformer architectures, the "residual stream" refers to the main pathway of data flow where feature representations are incrementally accumulated through skip connections (i.e., $x \leftarrow x + \text{Sublayer}(x)$).
>
> We mentioned it in the first paragraph to highlight that token merging directly intervenes in this primary pathway. If the aggregation breaks statistical conservation (as we prove it does), the resulting energy drift will propagate and compound layer-by-layer along this stream, destabilizing the deep feature representations. To improve the presentation, we will add a clear definition in the introduction of the final manuscript to ensure maximum clarity.
>
> **Q1: Hidden Constant Factors & The Actual Scaling Parameter Used**
> **Direct Answer:** In our experiments, exactly $\sqrt{0.75}$ is used for a retention ratio of $r=0.75$. **There are absolutely no hidden constant factors or empirical tuning parameters in our implementation.** *Explanation:* This parameter-free nature is one of the core strengths of our method. You astutely pointed out that our derivation relies on asymptotic approximations (e.g., $d \gg 1$). The reason we do not need to heuristically tune a hidden constant is driven by three key factors:
> 1. **Radial Correction (Full Dimension $D$):** Our radial scaling operates directly on the full residual stream ($D=384$ for DeiT-S, $D=768$ for ViT-B). At these scales, the "concentration of measure" bounds are remarkably tight, making our asymptotic derivation for energy conservation near-perfect.
> 2. **Angular Correction (Head Dimension $d$):** Our angular LMMSE correction operates within individual attention heads ($d=64$). While this is a finite-dimensional approximation compared to $D$, empirical dot product distributions in standard Transformers at $d=64$ remain sufficiently concentrated for our zero-mean cross-term assumption to hold robustly in practice without manual tuning.
> 3. **Architectural Constraints:** Normalization layers (like LayerNorm) explicitly enforce the stationary variance properties we assume, strictly preventing the theoretical-to-empirical gap from widening.
>
>
> ---
> **Conclusion**
> We are very grateful for your insightful questions, which allowed us to emphasize the parameter-free elegance and strict mathematical grounding of our method. Given that all reviewers have recognized the novelty, simplicity, and empirical effectiveness of our approach across broad benchmarks, *we kindly hope you might consider raising your score to reflect these strengths.* Thank you again for your time and invaluable feedback.

---

> > ### Author Rebuttal · Reviewer_Rqbm · 2026-04-03
> >
> > My concerns surrounding the presentation (W1) have been resolved. I will increase my score accordingly.

---

### Official Review · Reviewer_jX53 · 2026-03-16

**Soundness:** 3
**Presentation:** 4
**Significance:** 3
**Originality:** 2
**Overall Recommendation:** 5
**Confidence:** 4

**Summary:**

The authors present a study and propose methods for improved scaling with token merging schedules. At its core, the paper argues that token merging adds distortion by breaking core statistical assumptions that transformers rely on, which invokes a distribution shift in aggregated outputs, in terms of radial drift (magnitude) and angular drift (direction). The authors argue that previous works such as proportional attention corrects mostly for the mean of distributions, but does not provide enough correction for drift in variance.
From their study, the authors discover / propose the rule of $\sqrt{r}$ scaling for retention ratios $r$.

The authors empirically demonstrate that $\sqrt{r}$ scaling provides more stable performance with stronger compression schedules, and that their scaling method improves across different token merging strategies.

**Compliance With Llm Reviewing Policy:**

Affirmed.

**Final Justification:**

The authors meaningfully addressed my concerns, prompting an increase from a score of (4) to (5). Naturally, I expect the authors to address the revisions dutifully in a potential camera ready version.

**Key Questions For Authors:**

1. How does $\sqrt{r}$ scaling affect models in a setting where the model is trained from scratch with token merging?
2. Can the derived scaling law be applied in pruning methods as well as scaling methods? If not, why?
3. How does normalization layers (layer norm / RMSNorm) influence the properties of scaling? What about explicit QK normalization?
4. Can the scaling be said to be truly optimal, as claimed, or an approximation under reasonable assumptions?

As it stands, the central concern from this reviewer is a lack of discussions on potential limitations. The reviewer is willing to increase their score if discussions of limitations in the paper are made clear and explicit in a potential revision.

**Limitations:**

The conclusion focuses on the success of the statistical correction and empirical gains rather than discussing failure cases or constraints. Explicit mention of these would help readers understand how the method can be applied in practice, where the scaling law will hold, and where it might break. The closest thing to limitations appears indirectly in the assumptions section (finite second moments, variance stationarity enforced by normalisation), but these are presented as reasonable conditions rather than potential weaknesses. A more explicit treatment would be prudent to include for a final draft of the manuscript. Without these, it becomes much harder to assess generalisation beyond the evaluated settings, and particularly important for a theoretical work.

**Strengths And Weaknesses:**

### Strengths

1. The paper provides a clear, intuitive and compelling argument for the properties of token merging under different retention ratios.
2. The authors posit their method alongside recent research in token merging, and provides novel insights in their study.
3. The scaling method is simple and effective, and provides little overhead, and integrates well with existing strategies. The theoretical justification is an interesting read, and provides a reasonable hypothesis, validated with strong empirical evidence.

### Weaknesses

1. Some of the wording comes across as somewhat overpromising. "Statistically optimal" in the title is a strong claim.
    - Some of the assumptions on high-dimensional statistics is naturally simplified. There is an assumption of finite second moments and independently distributed noise that implies that the method is more of an approximation than a true optimal strategy. This is fine, but should be more explicitly emphasised in the manuscript.
    - Assumptions on optimal reconstruction seems to rely on predominantly linear assumptions, which might not hold in a nonlinear case, particularly for layer norm, see below.
2. There is limited discussion on the effect of layer norm in conjunction with the scaling rule. As this reviewer understands the method, the effect of normalization layers may have an effect on the results. This is also somewhat ommited from the theoretical analysis.
3. The paper lacks a dedicated section / discussion on potential limitations, see below for details.
4. (Minor) The $\sqrt{r}$ scaling rule is intuitive partly because it directly mirrors variance normalisation under reduced sample sizes. This can be interpreted as an insight that closely follows classical statistical principles, but also makes the claims of novelty slightly weaker.

### Justification of Scores

- Soundness (3, good): The theoretical analysis is sound and well suited for the arguments and claims made in the paper. Some slight issues are mentioned in weaknesses above.
- Presentation (4, excellent): The paper is well structured, easy to follow, and presents its findings clearly. Figures are clear and well constructed, aligning with the arguments made in the text.
- Significance (3, good): The findings are generalisable, and has the potential to affect token merging methods in vision in general.
- Originality (2, fair): As mentioned, the $\sqrt{r}$ finding essentially follows standard variance scaling arguments. Not a major drawback, in this reviewers opinion.

---

> ### Author Rebuttal · Authors · 2026-03-25
>
> We sincerely thank Reviewer jX53 for the encouraging review, the "Weak Accept" rating, and for recognizing the novelty and effectiveness of our method. We highly value your constructive feedback. Below is our point-by-point response.
>
> **W1, Q4:** Our original intent was to demonstrate mathematical optimality strictly under reasonable, ideal assumptions (finite second moments and variance stationarity). We acknowledge that real-world deep learning environments, with their inherent non-linearities and complex token correlations, deviate from these ideal conditions. Therefore, our method is more accurately described as an *asymptotically optimal approximation*. In the revised manuscript, we will explicitly scope our claims to prevent overpromising and to emphasize this classical statistical connection.
>
> **W2, W3, W4, Q3:** We appreciate you pointing out the need for a deeper discussion on non-linearities and normalization. As requested, we will add a dedicated **"Limitations and Operating Regimes"** section to explicitly bound our method's applicability:
> * **LayerNorm/RMSNorm (W3, Q3):** As detailed in our theoretical framework in Appendix A.1 (Assumption A.2), normalization is the critical architectural prior validating our "Variance Stationarity" assumption. Without it, aggregation-induced $1/r$ energy inflation would compound exponentially. LayerNorm resets the variance per block, meaning our $\sqrt{r}$ scaling acts as a vital *intra-block* geometric correction before the next normalization step.
> * **Explicit QK Normalization (W2, Q3):** Explicit QK normalization strictly bounds logit variance ($\sigma^2$), which inherently suppresses PA's estimation bias. In such variance-constrained architectures, the marginal utility of our Angular Correction (LMMSE logit shrinkage) will naturally decrease.
> * **Extreme Compression (W2):** Our LMMSE correction assumes intra-cluster noise is relatively independent. At extreme compression ratios, clusters group highly correlated semantic concepts, degrading this assumption. However, this theoretical boundary lies beyond the practical operating range where baseline accuracy is already severely degraded, making it empirically negligible in real-world deployments.
>
> **Q1:** If a model is trained from scratch with token merging, the optimizer natively absorbs variance shifts into the weights. However, our scaling provides critical safeguards for zero-shot higher compression during inference.
>
> To demonstrate this, we evaluated the official ToMe DeiT-S weights, trained from scratch (ToMe + PA, $c=13$ merged tokens per layer). Testing zero-shot higher compression rates during inference, we logically applied only our Radial Correction ($\sqrt{r}$ scaling to $V$), as the weights had already absorbed the PA bias.
>
> | Inference Merged Tokens ($c$) | Vanilla (None) | PA (Training Setting) | **Ours (PA + Radial $\sqrt{r}$)** |
> |:---:|:---:|:---:|:---:|
> | 13 | 77.5% | **79.4%** | 79.3% |
> | 16 | 75.9% | **79.1%** | **79.1%** |
> | 20 | 70.4% | 77.3% | **77.4%** |
> | 24 | 56.6% | 72.2% | **72.7%** |
> | 30 | 40.8% | 61.0% | **62.3%** |
>
> *Analysis:* This empirically validates our theoretical prediction: radial drift is driven by the mismatch between training and inference compression ratios. At the training rate ($c=13$), the model is inherently adapted, so our explicit correction yields negligible differences. As compression increases zero-shot ($c=24, 30$), unmitigated radial drift re-emerges. Our radial scaling successfully bounds this drift, yielding up to +1.3% accuracy gains (62.3% vs 61.0%) over the PA baseline.
>
> **Q2:** Yes, our scaling is partially applicable to pure pruning and highly applicable to hybrid methods, as validated in **Section 4.2** (Line 261):
> * **Pure Pruning (e.g., DCT):** Radial Correction ($\alpha = \sqrt{r}$) applies perfectly since energy drift depends on the global retention ratio $r$, not local clustering. We demonstrated this by recovering +6.4% accuracy over the Vanilla baseline at $r=0.75$. (Angular Correction is inapplicable, as cluster size $N_j$ is undefined when tokens are discarded).
> * **Hybrid Pruning+Merging (e.g., ToFu):** $\sqrt{r}$ scaling robustly corrects mixed variance drift in these complex hybrid paradigms, demonstrating our framework's broad applicability beyond standard bipartite merging.
>
> ---
> **Conclusion**
> We hope this explicit bounding of our operating regimes and the empirical validation of training-from-scratch dynamics thoroughly address your core concerns. We are committed to integrating these clarifications into the final manuscript. *We kindly hope you might consider raising your score.* Thank you for your invaluable feedback.

---

> > ### Author Rebuttal · Reviewer_jX53 · 2026-04-03
> >
> > The authors addressed concerns adequately. I believe my initial rating is accurate in relation to other papers, but as a score of (4) should be used sparingly, I increase my score to a (5) to reflect the clear and lucid response from the authors, provided they deliver on the stated (minor) revisions to the paper.
> >
> > I wish the authors luck in the final phase of the rebuttal.

---

> > > ### Author Response · Authors · 2026-04-03
> > >
> > > We sincerely thank you for the score upgrade and your encouraging words!
> > >
> > > Please rest assured that we will faithfully incorporate all the promised minor revisions into the final camera-ready manuscript, as the system does not permit PDF updates during this discussion phase.
> > >
> > > Thank you again for your invaluable guidance and for supporting our work!

---

### Decision · Program_Chairs · 2026-04-30

**Decision:**

Accept (regular)

**Comment:**

The reviewers agree that this paper makes a clear and practically useful contribution to token merging in Transformers by introducing a simple correction derived from a statistical reconstruction perspective. The main strengths are the clean radial/angular decomposition, the negligible computational overhead, and consistently positive empirical results across multiple token merging schemes and tasks. Concerns focused mainly on the strength of the “optimal” claim, the role of assumptions such as variance stationarity and high-dimensional approximations, and the need for a clearer discussion of limitations and operating regimes. The authors’ rebuttal addressed these points well: they scoped the claims more carefully, clarified the role of normalization, added discussion of limitations, and provided additional empirical justification. Reviewer sentiment is positive overall.